# A Phylogenetic Morphometric Investigation of Interspecific Relationships of *Lyponia* s. str. (Coleoptera, Lycidae) Based on Male Genitalia Shapes

**DOI:** 10.3390/insects15010011

**Published:** 2023-12-27

**Authors:** Chen Fang, Yuxia Yang, Xingke Yang, Haoyu Liu

**Affiliations:** 1Key Laboratory of Zoological Systematics and Application, School of Life Science, Institute of Life Science and Green Development, Hebei University, Baoding 071002, China; 20228017069@stumail.hbu.edu.cn; 2Key Laboratory of Zoological Systematics and Evolution, Institute of Zoology, Chinese Academy of Sciences, Beijing 100101, China; yangxk@ioz.ac.cn

**Keywords:** *Lyponia*, taxonomy, morphology, net-winged beetles, Oriental region

## Abstract

**Simple Summary:**

Resolving phylogenetic relationships among animals remains one of the most challenging issues in systematics. Currently, molecular phylogeny is the standard for inferring evolutionary relationships, but morphological analysis still cannot be replaced or neglected. Male genitalia have been proven to be valuable in phylogenetic analyses usually in the higher taxonomic grades but are rarely studied at the lower level. In the present study, we performed a taxonomic review (with two new species described) and further investigated the interspecific relationship of *Lyponia* s. str. based on the morphometric data of phallus shapes using geometric morphometric (GM) and phylogenetic morphometric (PM) methods. As a result, the produced topologies (of the unweighted pair group method using arithmetic averages (UPGMA), neighbor-joining (NJ) and maximum parsimony (MP) analyses) provide a general framework of the morphological evolution of this subgenus. To be exact, these species are divided into two clades that represent two shapes of the phallus. The results provide better understanding of the species diversity and evolution of *Lyponia* s. str. and shed new light on investigations of the phylogenetic relationships of insects based on male genitalia shapes, which is particularly useful when molecular data are unavailable.

**Abstract:**

The nominate subgenus *Lyponia* Waterhouse, 1878 from China is reviewed, with two new species described and named *L.* (s. str.) *ruficeps* sp. n. (China, Yunnan) and *L.* (s. str.) *zayuana* sp. n. (China, Xizang). A distribution map and a key to all species of *Lyponia* s. str. are provided. Moreover, the phenotypic relationships among the species of *Lyponia* s. str. are investigated based on phallus shapes using geometric morphometric and phylogenetic morphometric analyses. The topologies demonstrate that the species are divided into two clades. One clade is composed of six species (*L. ruficeps* sp. n., *L. zayuana* sp. n., *L. kuatunensis*, *L. shaanxiensis*, *L. hainanensis*, and *L. tamdaoensis*) and is supported by a stout phallus (less than 3.6 times longer than wide). The other clade includes the remaining species (*L. nepalensis*, *L. debilis*, *L. cangshanica*, *L. delicatula*, and *L. oswai*) and is supported by a slender phallus (at least 4.1 times longer than wide). These results provide better understanding of the species diversity and evolution of *Lyponia* s. str. Nonetheless, more samples and loci are required in the future to verify the present results.

## 1. Introduction

Insecta is the largest organism group on the earth, with a high diversity of morphological variations [1]. Male genitalia of insects are one of the most evolutionarily variable morphological characters, and their apparently rapid rate of morphological changes is assumed to result from sexual selection [2]. They are helpful for allowing systematists to identify diagnostic features at various taxonomic levels [3]. As noted in other insects, male genitalia are the most important and useful diagnostic characteristics of species of net-winged beetles, including *Lyponia* [4]. They have also been demonstrated to be useful in phylogenetic analyses [5] and are usually applied in higher-level classifications and phylogenetic frameworks [6,7,8]. Although there are few studies assessing the phylogenetic value of male genitalia in the lower grades due to the uniformity of their structures, it has become possible to elucidate the phenotypic relationships among the species with the advent of geometric morphometric (GM) [9] and phylogenetic morphometric (PM) methods [10]. Following our preceding publication [11], herein we construct the phenotypic relationships of *Lyponia* s. str. based on male genitalia shapes. Although Bocak [12] conducted a cladistics analysis of *Lyponia*, he noted that the analysis was only for the species groups not the species given that the high morphological similarity in some species preventing coding the dataset. More recently, Li et al. [13] proposed a phylogenetic hypothesis for Lyponiini based on molecular data, in which only a few species of *Lyponia* s. str. were included in the analysis because of limited availability of the material. Therefore, it is necessary for us to comprehensively assess the phylogenetic relationships among the species of *Lyponia* s. str. herein.

The genus *Lyponia* was proposed by Waterhouse for *L. debilis* Waterhouse, 1878 [14]. It was always mixed with *Ponyalis* Fairmaire, 1899 [15] due to their similarity in general appearances [15,16,17,18,19]. For this reason, Nakane [20] created a broad concept of *Lyponia* s. l. that included both *Ponyalis* and *Lyponia* s. str. Thirty years later, *Lyponia* s. l. was systematically revised by Bocak [12], in which *Ponyalis* was treated as a subgenus of *Lyponia*, and another subgenus *Weiyangia* was proposed. Soon after that, Kazantsev reinstated *Ponyalis* as a separate genus and subdivided *Lyponia* into five subgenera: *Lyponia* s. str.; *Weiyangia* Bocak, 1999; *Poniella* Kazantsev, 2002 (the *L. nigroscutellaris* group by Bocak, 1999); *Mimoditonecia* Kazantsev, 2002; and *Sundolyponia* Kazantsev, 2002 [21]. Regardless of the classification system, *Lyponia* is most closely related to *Ponyalis*, and they belong to the subtribe Lyponiinina [7] or the tribe Lyponiini [8,22,23,24] based on either morphological or molecular evidence. The members of *Lyponia* are distinguished from those of *Ponyalis* based on the following characters: free basal part of the coxite, while basally fused in *Ponyalis*; antennomere I of both sexes progressively widened and not compressed anteriorly, while abruptly widened near the base and more or less compressed at the anterior margin in *Ponyalis*; primary elytral costa III not going beyond the apical fifth, while nearly reaching the apex of the elytra in *Ponyalis*; aedeagus absent with numerous minute thorns in the preapical portion of the median lobe, while present in *Ponyalis* [21].

At present, a total of 27 species of *Lyponia* sensu Kazantsev are hitherto known [4,12,14,15,16,17,18,19,20,21,25,26,27,28,29,30,31], and they are widely distributed in the eastern Palaearctic and Oriental regions [4,12,21]. Among the subgenera, *Lyponia* s. str. is easily distinguished from others based on the larger eyes and male antennomere III equipped with slender lamella [21]. Prior to this study, this subgenus contained nine species [4,12,14,21,25,29,30], of which five species are present in China. Recently, we obtain some specimens of *Lyponia* s. str. from China. After our examination and identification, we discover two new species and a species newly recorded in China that are reported in the present study. Meanwhile, the previously known species are illustrated to make comparisons with the new species. Moreover, the phylogeny of *Lyponia* s. str. is constructed to assess the phonetic relationships among the species. The results of this study will allow us to better understand the species diversity and evolution of *Lyponia* s. str.

## 2. Materials and Methods

### 2.1. Material

The studied material is preserved in the following collections: IZAS (Institute of Zoology, Chinese Academy of Sciences, Beijing, China) and MHBU (Museum of Hebei University, Baoding, China).

### 2.2. Morphological Techniques

We softened the dry specimens with water, then dissected the aedeagus, and cleared it in 10% NaOH solution. After that, we examined and took photos of the aedeagus, which was immersed in glycerol, and then glued it on a paper card for permanent preservation. We obtained images of the habitus with a Canon EOS 80D digital camera (Canon (China) Co., Ltd., Zhuhai, China) and aedeagi using a Leica M205A stereomicroscope (Leica Microsystems Inc., Heerbrugg, Switzerland). Then, we stacked images of different layers using Helicon Focus 7. In the end, we edited the final permutation in Adobe Photoshop CS3.10.0.1.

We performed the following measurements using Image J 1.50i (NIH, Bethesda, MD, USA): body length (from the anterior margin of head to the elytral apex), body width (the width across the humeri of elytra), pronotal length (from middle of anterior margin to middle of posterior margin of pronotum), pronotal width (the width across the widest part of pronotum), eye diameter (the maximal width of an eye), and interocular distance (the minimal distance between eyes).

### 2.3. Map Preparation

We collected the distribution information from the original publications [2,3,4,6] and the material in this study. Based on the retrieved data, we prepared the distribution map using ArcMap 10.8 and edited it in Adobe Photoshop CS3.10.0.1.

### 2.4. Geometric Morphometric (GM) Analyses

#### 2.4.1. Data Collection

The phallus shapes of all *Lyponia* s. str. species were included in the analyses. The data of most species were collected from the material in this study, and data from a few samples were obtained from previous publications [12,21] (Table 1).

#### 2.4.2. Data Acquisition and Digitalization

We digitized the photographs or scanned images of aedeagi using tps-DIG 2.12 software [32]. For each phallus, a curve was extracted from the dorsal contour to present the external form. The curve started from the base and ended at the same point and was resampled into a total of 150 equally spaced semi-landmarks (Figure 1). To avoid landmark measurement error, the same observer (C. Fang) repeated the digitalization procedure three times on different days.

Then, we input the morphometric data into the tps-UTIL 1.43 software [33] and used TpsSmall (ver. 1.20, F. Rohlf, see http://life2.bio.sunysb.edu/ morph, assessed on 10 September 2023) to test whether the observed variation in shape was small enough that the distribution of points in the tangent space can be used as a good approximation of the distribution in shape space. We analyzed the coordinates using TpsRelw 1.46 software [34] to calculate the Eigen values for each principal warp. Finally, we showed the shape changes of different species as thin-plate splines based on variation along the first two relative warp axes.

#### 2.4.3. Data Analyses

We analyzed the aligned landmarks data (which was converted from semi-landmarks in the tps. file manually) using MorphoJ 1.06d software [35] to examine the shape variation. We employed principal component analysis (PCA) to test how well the species can be separated based on the shape of the phallus. The observed variation patterns among the species frequently corresponded to the characters with high loading value in PCAs. We used canonical variates analysis (CVA) to analyze the relative similarity and discrimination of the test groups. The shape values that maximize group means relative to variation within groups were determined using CVA, in which the covariate matrices are assumed to be identical [36]. We computed the Procrustes distances and Mahalanobis distances (the square root of the sum of squared differences between corresponding points) between each of the species and produced the matrices using MorphoJ software [35].

### 2.5. Phylogenetic Morphometric (PM) Analyses

We investigated phylogenetic relationships among the species of *Lyponia* s. str. based on the morphometric data of male genitalia using unweighted pair group method using arithmetic averages (UPGMA), neighbor-joining (NJ) and maximum parsimony (MP) analyses [37,38].

We conducted UPGMA and cluster analyses based on the Procrustes and Mahalanobis distances matrices (Appendix A). Then, we loaded the Procrustes and Mahalanobis distance score matrices in PAST 2.17 [39], separately, to assess the phonetic relationships among the species.

In addition, we constructed NJ trees [40] using PAST 2.17 with 1000 bootstrap replicates to display the Mahalanobis and Procrustes distances between populations, separately.

Moreover, we produced the tps files in tps-DIG (Appendix A) to perform MP analysis using TNT 1.5 [38]. We followed a heuristic (traditional search) search strategy. Here, random addition sequences were utilized, and tree bisection reconnection (TBR) was used as a branch swapping algorithm. In each replicate, a separate tree was obtained, and the runs were repeated 1000 times (mult = ras tbr hold 1 rep 1000) [10].

## 3. Results

### 3.1. Taxonomy


**Genus *Lyponia* Waterhouse, 1878**


Diagnosis. Body black to dark-brown, small to medium sized (5.0–15.0 mm in length). Head small, with hemispherically prominent eyes. Antennal 11-segmented, filiform in both sexes, or flabellate in males while serrate in females. Male antennomere III with short or long lamella, or flattened setose exterior. Pronotum red, always mixed with dark or brown patch on disc, square or trapezoidal, without conspicuous pronotal carinae. Scutellum narrowed posteriorly, with apical margin widely rounded or shallowly emarginate in middle. Elytra uniformly red, subparallel-sided, cells variable in shape. Aedeagus asymmetric, without parameres, phallus slender to stout, internal sac generally developed.




**Key to the subgenera of *Lyponia* sensu Kazantsev (2002)**


1.Male antennomere III with slender lamella………………………………………………2-Male antennomere III without any lamella……………………………………………….32.Eyes small and interocular distance at least 1.7 times as wide as their diameter………………………………………………………………………*Lyponia* s. str.-Eyes large and interocular distance at most 1.3 times as wide as their diameter………………………………………………………………*L*. (*Mimoditonecia*) Kazantsev3.Male antennae feebly serrate, antennomere III unmodified………………………………4-Male antennae pectinate, antennomere III with flattened setose exterior or lamella………………………………………………………………………*L*. (*Poniella*) Kazantsev4.Primary elytral costae easily separable from secondary ones; male trochanters simple………………………………………………………………………*L*. (*Weiyangia*) Bocak-Primary elytral costae indistinguishable from secondary ones, except at base; male trochanters spinose………………………………………………*L*. (*Sundolyponia*) Kazantsev




**Subgenus *Lyponia* Waterhouse, 1878**


Included species. *L. cangshanica* Li, Pang & Bocak, 2015 [4]; *L. debilis* Waterhouse, 1878 [14]; *L. delicatula* (Kiesenwetter, 1874) [25]; *L. hainanensis* Li, Pang & Bocak, 2015 [4]; *L. kuatunensis* Bocak, 1999 [12]; *L. nepalensis* Nakane, 1983 [29]; *L. osawai* Nakane, 1969 [20]; *L. shaanxiensis* Kazantsev, 2002 [21]; *L. tamdaoensis* Kazantsev, 2002 [21]; *L. ruficeps* sp. n.; and *L. zayuana* sp. n.

Distribution (Figure 2). China (Anhui, Fujian, Hubei, Hainan, Jiangxi, Guangdong, Guangxi, Guizhou, Yunnan, Zhejiang, Xizang), Japan, Nepal, and Vietnam.


***Lyponia* (s. str.) *cangshanica* Li, Pang & Bocak, 2015**


Figure 2 and Figure 3A–C

*Lyponia* (s. str.) *cangshanica* Li, Pang & Bocak, 2015 [4]: 12, Figures 1 and 9.

Material examined. China—1♂ (IZAS), Yunnan, Deqin, Baimangxueshan, 3300 m, 28.VII.1987, S. Y. Wang leg.

Descriptive notes. Male. Antennae flabellate, overlapping basal two-third length of elytra when inclined. Antennomere II transverse; III–XI lamellate, lamella of IV with nearly as long as the joint itself, and lamella of IX 1.8 times as long as the joint itself.

Aedeagus (Figure 3A–C): Phallus stout and approximately 3.0 times as long as wide, narrowed at the apical part and tapered at the apex in ventral and dorsal views (Figure 4A,B), distinctly bent dorsally in the lateral view (Figure 3C), internal sac reduced and minimally visible (Figure 3A–C).

Distribution (Figure 2). China: Yunnan.

Remarks. In the original publication [4], the male habitus was well illustrated, but the aedeagus was only described with illustrations in the ventral view. Here, we provided macrophotographs of the aedeagus in ventral, dorsal and lateral views for this species to make it better understood. In addition, we described the male antennae in more detail to make comparisons with other species, since this character is vital for specific identification.


***Lyponia* (s. str.) *debilis* Waterhouse, 1878**


Figure 2, Figure 3D–F and Figure 4A,B

*Lyponia debilis* Waterhouse, 1878 [14]: 107; Bocak [12]: 78, Figure 61; Kazantsev [21]: 199; Li et al. [4]: 15.

*Lyponia pieli* Pic, 1937: 169. Synonymized by Bocak [12]: 78.

Material examined. China—1♂1♀ (IZAS), Anhui, Huoshan, Shimozitang, Huangnibao, 902.16 m, 5.V.2021, K. D. Zhao leg.; 1♂ (IZAS), Anhui, Yuexi, Yaoluoping, 1057.97 m, 16.V.2021, K. D. Zhao & X. C. Zhu leg.; 1♂1♀ (MHBU), Anhui, Yuexi, Yaoluoping, VII.2015, J. Fang leg.; 2♂2♀ (MHBU), Zhejiang, Qingliangfeng, Tianchi, 21.V.2012, J. S. Xu & L. X. Chan leg.; 2♂2♀ (IZAS), Zhejiang, Anji, Longwangshan, 13.V.1996, H. Wu leg.; and 1♂ (IZAS), Zhejiang, Qingyuan, Xishanzu, 18.IV.1994, H. Wu leg.

Descriptive notes. Male (Figure 4A). Antennae flabellate, overlapping two-third length of elytra when inclined. Antennomere II transverse; III–XI lamellate, lamella of IV ca. 1.7 times as long as the joint itself, and lamella of IX 1.6 times longer than the joint itself.

Aedeagus (Figure 3D–F): phallus stout and 2.6 times as long as wide, widened near middle part and tapered at apex in dorsal and ventral views (Figure 3D,E), nearly straight in the lateral view (Figure 3F), internal sac robust and obviously protruding over the apex of the phallus (Figure 3D–F).

Female (Figure 4B). Similar to male, but body stouter, antennae serrate.

Distribution (Figure 2). China (Anhui, Fujian, Jiangxi, Zhejiang, Guangdong, Guangxi, Guizhou, Hunan).

Remarks. Although the male of this species has been mentioned several times [4,12,21], its aedeagus was illustrated in the ventral view only by Bocak [12]. In the present study, the habitus of the male and female for this species are provided for the first time, and the aedeagus is illustrated in ventral, dorsal and lateral views to provide more information.




***Lyponia (s. str.) kuatunensis* Bocak, 1999**


Figure 2, Figure 3G–I and Figure 4C,D

*Lyponia* (s. str.) *kuatunensis* Bocak, 1999 [12]: 79, Figures 31 and 64; Kazantsev [21]: 199; Li et al. [4]: 15.

**Material examined.** China—5♂7♀ (MHBU), Guangxi, Wuming, Damingshan, 1100 m, 27.V.2011, H. Y. Liu leg.

**Descriptive notes.** Male (Figure 4C). Antennae flabellate, overlapping the midlength of elytra when inclined. Antennomere II transverse; III–XI lamellate, and lamella of IV ca. 1.2 times longer than the joint itself, and lamella of IX 1.6 times as long as the joint itself.

Aedeagus (Figure 3G–I): phallus moderately slender and about 3.5 times as long as wide, feebly widened apically and tapered at apex in dorsal and ventral views (Figure 3G,H), feebly bent dorsally in lateral view (Figure 3I), internal sac robust, but never protruding over the apex of the phallus (Figure 3G–I).

Female (Figure 4D). Similar to male, but body stouter, antennae serrate.

Distribution (Figure 2). China (Fujian, Guangxi, Hunan, Guizhou).

Remarks. Bocak [12] provided the illustrations of male habitus and aedeagus in ventral view for *L. kuatunensis*. In this study, the habitus of both sexes, and the aedeagus in ventral, dorsal and lateral views are illustrated to provide more morphological details.




**
*Lyponia (s. str.) hainanensis*
**
**Li, Bocak & Pang, 2015**


Figure 2 and Figure 3J–L

*Lyponia* (s. str.) *hainanensis* Li, Bocak & Pang, 2015 [4]: 11, Figures 2 and 10.

Material examined. China—1♂ (IZAS), Guangxi, Maoershan, 8.VII.1985, S. M. Song leg.

Descriptive notes. Male. Antennae flabellate, overlapping the midlength of elytra when inclined. Antennomere II transverse; III–XI lamellate, and lamella of IV ca. 1.7 times as long as the joint itself, and lamella of IX 1.8 times longer than the joint itself.

Aedeagus (Figure 3J–L): phallus slender and approximately 4.1 times as long as wide, widened apically and tapered at apex in dorsal and ventral views (Figure 3J,K), feebly bent dorsally in the lateral view (Figure 3L), internal sac well developed and protruding over the apex of the phallus (Figure 3J–L).

Distribution (Figure 2). China (Hainan, Guangxi).

Remarks. In the original description provided, the photos of the male habitus and aedeagus were in the ventral view [12]. In this study, we illustrate the aedeagus in ventral, dorsal and lateral views to make easier to compare with others.




***Lyponia (s. str.) shaanxiensis* Kazantsev, 2002**


Figure 2, Figure 5A,B and Figure 6A–C

*Lyponia shaanxiensis* Kazantsev, 2002 [21]: 199, Figures 15 and 16; Li et al. [4]: 15.

Material examined. China—1♂1♀ (IZAS), Shaanxi, Ningshan, Huoditang, 800–1580 m, 15.VII.1998, D. C. Yuan leg.; 2♂1♀ (MHBU), Henan, Songxian, Baiyunshan, 14.-17.VII.2008, G. D. Ren & Q. Q. Wu leg.; 1♂ (MHBU), Henan, Xixia, Huashuapang, 19. VII.2008, G. D. Ren & Q. Q. Wu leg.

Descriptive notes. Male (Figure 5A). Antennae flabellate, overlapping basal two-thirds length of elytra when inclined. Antennomere II transverse; III–XI lamellate, lamella of IV ca. 1.9 times as long as the joint itself, and lamella of IX 3.0 times as long as the joint itself.

Aedeagus (Figure 6A–C): phallus slender and about 3.6 times as long as wide, widened apically and rounded at apex in ventral and dorsal views (Figure 6A,B), “S”-shaped in lateral view (Figure 6C), internal sac slender and slightly protruding over the apex of the phallus (Figure 6A–C).

Female (Figure 5B). Similar to male, but more slender than male, and antennae serrate.

Distribution (Figure 2). China (Henan, Shaanxi).

Remarks. In the present study, the female is firstly reported, and the habitus of both sexes are provided for the first time. In addition, we discover that the body coloration of this species is somewhat variable, and the individuals from Shaanxi always have a black patch on the pronotum and black scutellum (Figure 5B), while those from Henan have a dark-brown patch on the pronotum and light red scutellum (Figure 5A).




***Lyponia (s. str) nepalensis* Nakane, 1983**


Figure 2, Figure 5C and Figure 6D–F

*Lyponia nepalensis* Nakane, 1983 [29]: 116, Figures 1–3; Bocak [12]: 75; Kazantsev [21]: 199; Li et al. [4]: 15.

Material examined. China—1♂ (IZAS), Xizang, Nyalam, Zham, 2250 m, 10.V.1974, X. Z. Zhang leg.

Descriptive notes. Male (Figure 5C). Antennae flabellate, overlapping basal two-thirds length of elytra when inclined. Antennomere II transverse; III–XI lamellate, lamella of IV ca. 1.5 times as long as the joint itself, and lamella of IX twice as long as the joint itself.

Aedeagus (Figure 6D–F): phallus stout and 3.0 times as long as wide, widened to one side and widely rounded at the apex in dorsal and ventral views (Figure 6D,E), nearly straight in the lateral view (Figure 6F), internal sac reduced and minimally visible (Figure 6D–F).

Distribution (Figure 2). China (new record: Xizang); Nepal.

Remarks. The species is recorded to China for the first time.




***Lyponia (s. str.) zayuana* Liu, Fang & Y. Yang, sp. n.**


Figure 2, Figure 5D and Figure 6G–I

Type material. Holotype, China—♂ (IZAS), Xizang, Zayu, Cawarong, 3392 m, 15.IX.2014, H. B. Liang leg.

Differential diagnosis. It is most similar to *L. nepalensis* on basis of the general appearance but can be easily distinguished from the latter based on the following characters: lamella of antennomere IX slender in male, about 10.0 times longer than wide (Figure 5D), while robust in *L. nepalensis*, about 6.0 times longer than wide (Figure 5C); pronotum with posterior margin arched (Figure 5D), while nearly straight in *L. nepalensis* (Figure 5C); phallus nearly parallel-sided with a robust internal sac (Figure 6G–I), while widened to one side with feebly visible internal sac in *L. nepalensis* (Figure 6D–F).

Description. Male. (Figure 5D). Length 10.3 mm, width at humeri 2.3 mm. Body black; pronotum, elytra, and scutellum red.

Head dorsally flat, antennal tubercles and transversal depression present; antennae almost reaching the basal two-thirds length of the elytra, antennomere II transverse; III–XI with short and thin lamellae, lamella of IV 1.4 times as long as the joint itself and 1.7 times as long as wide, lamella of IX 1.8 times longer than the joint itself and 10.0 times as long as wide.

Pronotum almost trapezoidal, with rounded anterior angles and acute posterior angles, anterior margin widely rounded and slightly producing anteriorly and posterior margin strongly arched posteriorly. Scutellum slightly narrowed posteriorly and straight at the apex.

Elytra slightly widened posteriorly, only primary costae Ⅱ and VI stouter than secondary ones, secondary costae III shortened to the apical fifth length, cells irregular at the basal part of the elytra.

Aedeagus (Figure 6G–I): phallus moderately slender and 3.5 times as long as wide, nearly parallel-sided, narrowly rounded at apex in dorsal and ventral views (Figure 6G,H), feebly bent dorsally in the lateral view (Figure 6I), internal sac moderately developed and robust, but never protruding over the apex of the phallus (Figure 6G–I).

Female. Unknown.

Distribution (Figure 2). China (Xizang).

Etymology. The name of the species is derived from the name of the type locality “Zayu”, Xizang Autonomous Region, China.




***Lyponia (s. str.) ruficeps* Liu, Fang & Yang, sp. n.**


Figure 2, Figure 5E,F and Figure 6J–L

Type material. Holotype, China—♂ (MHBU), Yunnan, Baoshan, Gaoligongshan, 2250 m, 21.–23.VI.2011, Y. X. Yang leg. Paratypes, 3♂2♀ (MHBU), same data as the holotype.

Differential diagnosis. It looks similar to *L. kuatunensis* in general appearance, but can be differentiated by the uniformly red pronotum (Figure 5E,F), while a dark-brown patch is always present in *L. kuatunensis* (Figure 4C,D); lamella of antennomere IX slender in male, about 15.0 times longer than wide (Figure 5E), while much shorter in *L. kuatunensis*, about 7.0 times longer than wide (Figure 4C); phallus subparallel-sided in ventral view (Figure 6J), while progressively widened apically in *L. kuatunensis* (Figure 3G).

Description. Male. (Figure 5E). Length 6.5–7.5 mm, width at humeri 1.5–1.8 mm. Body black; pronotum, elytra, and scutellum red.

Head dorsally flat, antennal tubercles and transversal depression present; antennae nearly reaching middle length of the elytra, antennomere II transverse; III–XI with short and thin lamellae, lamella of IV 1.4 times as long as the joint itself and 2.0 times as long as wide, lamella of IX 1.8 times longer than the joint itself and 15.0 times as long as wide.

Pronotum almost trapezoidal, with rounded anterior angles and rectangular posterior angles, anterior margin widely rounded and projecting anteriad, lateral margin sinuates and posterior margin bisinuate. Scutellum feebly narrowed posteriorly and obviously emarginate at the apex.

Elytra parallel-sided, all primary costae stouter than secondary ones, cell irregular, primary costa I and secondary costa I merged at apical part of disc, primary costa II stouter than all other costae.

Aedeagus (Figure 6J–L): phallus slender and about 4.5 times as long as wide, nearly parallel-sided and tapered at apex in dorsal and ventral views (Figure 6J,K), feebly bent dorsally in lateral view (Figure 6L), internal sac moderately developed and stout, but never protruding over apex of phallus (Figure 6J–L).

Female (Figure 5F). Stouter than male, antennal serrate, reaching nearly the midlength of elytra, pronotum with anterior angles confluent with anterior margin, scutellum feebly emarginate at the apex.

Distribution (Figure 2). China (Yunnan).

Etymology. The specific name is derived from the Latin “*rufus*” (red) +“-*ceps*” (head), referring to its uniformly red head.

### 3.2. GM Analyses of Male Genitalia Shapes

The correlation (uncentered) between the tangent space (Y) regressed onto the Procrustes distance (geodesic distances in radians) was 0.999997. There was little doubt on the basis of the result from tps-SMALL, which supported that the data are acceptable by geometric morphometric methods, since the results from the statistical test performed using tps-SMALL proved the acceptability of the data set for further statistical analysis [41].

The first three principal components of the shape of phallus explain 94.433% of the micromesh variation and were 68.041%, 21.877% and 4.525%, respectively (see Appendix A). They were plotted to indicate variation along the first two relative warp axes (Figure 7). The shape changes of all species of *Lyponia* s. str. were shown as deformations of the least squares reference using thin-plate splines (Figure 7A–L).

Comparison of the tps configurations indicated that the average shape of the phallus of *Lyponia* s. str. is asymmetrical and feebly bent to one side, almost even in width except feebly narrowed at the apical one-seventh, and rounded at apex in dorsal view (Figure 7A).

The first principal component (*x*-axis) dominated the uniformity of the phallus shape, and the species on the left seemed narrowed apically (e.g., Figure 7B,C,I–L). In contrast, those on the right appeared widened near apex (e.g., Figure 7D–G). Meanwhile, the second principal component (*y*-axis) decided the width of the phallus. The species below were generally stout (e.g., Figure 7H–L), while those above seemed slender (e.g., Figure 7B–H).

Moreover, the CVA and PCA scatter plots of shape differences of the phallus shapes in dorsal views (see Appendix A) showed that each species of *Lyponia* s. str. independently occupied an area and were separated from one another.

### 3.3. PM Analyses Based on Male Genitalia Shapes

The UPGMA phenogram based on Procrustes distances of phallus shapes in dorsal view (Figure 8A) indicated that the species of *Lyponia* s. str. were divided into two clades. One (clade I, Figure 8a) was composed of ((*L. hainanensis* + *L. tamdaoensis*) + *L. ruficeps* sp. n.) + (*L. shaanxiensis* + (*L. kuatunensis* + *L. zayuana* sp. n.)), and the other (clade II, Figure 8b) included ((*L. nepalensis* + *L. cangshanica*) + *L. oswai*) + (*L. delicatula* + *L. debilis*). Clade I was partially recovered (*L. hainanensis* + *L. tamdaoensis*, Figure 8c) in the UPGMA phenogram based on the Mahalanobis distance (Figure 8B). In contrast, clade II was fully recovered, although some changes within the clade were shown in the interrelationships among the species.

In the NJ tree of Procrustes distances (Figure 8C), clade I was recovered, but the interrelationships among the species were different from the above (Figure 8A). Although the species of clade II were not clustered together, the two-component small branches (*L. nepalensis* + *L. oswai*) + *L. cangshanica* (Figure 8e) and *L. delicatula* + *L. debilis* (Figure 8g) were recovered, separately. In comparison, both (*L. hainanensis* + *L. tamdaoensis*) of clade I and ((*L. nepalensis* + *L. cangshanica*) + *L. oswai*) of clade II were only partially recovered in the NJ tree of Mahalanobis distance (Figure 8C).

The MP tree of two landmark configurations (Figure 8E) produced a significantly different topology from the above phenograms. Only the sister relationship of *L. delicatula* + *L. debilis* of clade II was recovered. Although the coupled clade of *L. nepalensis* + *L. oswai* (Figure 8f) was different from that shown in clade I (Figure 8A,B,D,d), it was similar to that of the NJ tree of Mahalanobis distance (Figure 8C).

## 4. Discussion

### 4.1. Separate Status of the New Species

Like other net-winged beetles, reliable identification of the species is regularly based on the shape of the male genitalia [4]. Within the subgenus *Lyponia* s. str., the morphological interspecific divergence of male genitalia always exhibit quantitative variations [4,12,21]. Geometric morphometrics is a rigorous method [42,43] to identify and analyze shape variations between species [44].

In the present study, the phallus shapes of *Lyponia* s. str. were analyzed using GM analyses. The statistical test performed by TpsSmall suggested that our obtained data on male genitalia are acceptable for geometric morphometric analysis. Further, CVA analysis suggested that all species of *Lyponia* s. str. can be distinguished from one another based on male genitalia shapes. Moreover, the tps configurations effectively display the shape of the phallus for each species (Figure 7A–L), so it could be included in the comparison between sibling species and complied in the key, thereby facilitating and stabilizing the taxonomy of *Lyponia*. Also, a logical conclusion drawn from the observations is that male genitalia have always been useful as a taxonomic character for the species, which suggests that they must obtain a new form in every new species [45]. This hypothesis was validated in our study, and the two new species discovered here, *L. zayuana* sp. n. and *L. ruficeps* sp. n., have phallus shapes that differ from the previously known species.

In addition, the shape of male antennae is rarely used as a conspicuous variable within species [12,21], so it is also applied in the identification key for the species.




**Key to the species of *Lyponia* s. str.**


1.Lamella of male antennomere IX extremely long, at least 2.3 times longer than the joint itself (e.g., Figure 4A and Figure 5A) ………………………………………………………2-Lamella of male antennomere IX moderately long, at most twice longer than the joint itself (e.g., Figure 4C)………………………………………………………………………42.Lamella of male antennomere IX ca. 3.0 times longer than the joint itself; aedeagus: phallus ca. 3.6 times as long as wide, widest at apical part, and rounded at the apex in the dorsal view (Figure 6B and Figure 7G)……………*L.* (s. str.) *shaanxiensis* Kazantsev, 2002-Lamella of male antennomere IX less than 2.6 times longer than the joint itself; aedeagus: phallus ca. 2.5 times as long as wide, widest near the middle, and tapered at the apex in the dorsal view……………………………………………………………………………33.Lamella of male antennomere IX ca. 2.3 times longer than the joint itself; aedeagus: internal sac thin and slightly protruding over apex of the phallus ([12]: Figure 63) ………………………………………………………………*L.* (s. str.) *osawai* Nakane, 1969-Lamella of male antennomere IX ca. 2.6 times longer than the joint itself; aedeagus: internal sac robust and obviously protruding over apex of phallus (Figure 3D–F) …………………………………………………………*L.* (s. str) *debilis* Waterhouse, 18784.Aedeagus: phallus slender and at least 4.1 times as long as wide in the dorsal view (e.g., Figure 6J–L and Figure 7B) …………………………………………………………………5-Aedeagus: phallus stout and at most 3.5 times as long as wide in the dorsal view (e.g., Figure 3A–C and Figure 7H) ………………………………………………………………………75.Pronotum uniformly red (Figure 5E,F); aedeagus: internal sac moderately developed and robust, but never protruding over the apex of the phallus (Figure 6J–L and Figure 7B) ………………………………………………………………………*L.* (s. str.) *ruficeps* sp. n.-Pronotum red or orange yellow, with a black or dark-brown patch in center of disc (e.g., Figure 5D); aedeagus: internal sac well developed and thin, protruding over the apex of the phallus.……………………………………………………………………………66.Male antennomere IV nearly as long as the joint itself; elytra with primary costae III disappearing at basal third part; aedeagus: phallus ca. 4.6 times as long as wide in the dorsal view (Figure 7E and [4]: Figure 18)………*L.* (s. str.) *tamdaoensis* Kazantsev, 2002-Male antennomere IV about 1.6 times longer than the joint itself; elytra with primary costae III well developed in whole length; aedeagus: phallus ca. 4.1 times as long as wide in the dorsal view (Figure 3K and Figure 7D)…………………………………………………………………………………………………*L.* (s. str.) *hainanensis* Li, Bocak & Pang, 20157.Aedeagus: internal sac reduced and feebly visible (e.g., Figure 3A–C and Figure 6D–F)……8-Aedeagus: internal sac moderately developed and visible (e.g., Figure 6A–C)………98.Elytra with all primary costae stouter than secondary ones ([9]: Figure 1); aedeagus: phallus tapered at the apex in ventral and dorsal views (Figure 3A,B and Figure 7H) ……………………………………………*L.* (s. str.) *cangshanica* Li, Bocak & Pang, 2015-Elytra with only primary costae II and IV stouter than secondary ones (Figure 5C); aedeagus: phallus rounded at the apex in dorsal and ventral views (Figure 6D,E and Figure 7J) …………………………………………………………*L.* (s. str.) *nepalensis* Nakane, 19839.Elytra with secondary costae III and IV reduced, only reaching approximately the basal third length (Figure 4C,D); aedeagus: phallus ca. 3.5 times as long as wide, tapered at apex in the dorsal view (Figure 3G,H and Figure 7F) ……………………………………………………………………………………………*L.* (s. str.) *kuatunensis* Bocak, 1999-Elytra with secondary costae III and IV moderately developed, reaching at least midlength; aedeagus unlike above………………………………………………………1010.Elytra with secondary costae III and IV at most extending to midlength; aedeagus: phallus ca. 2.9 times as long as wide, tapered at apex in the dorsal view (Figure 7K and [12]: Figure 66)……………………………*L.* (s. str.) *delicatula* (Kiesenwetter, 1874)-Elytra with secondary costae III and IV reaching apical fifth length (Figure 5D); aedeagus: phallus ca. 3.5 times as long as wide, narrowly rounded at apex in the dorsal view (Figure 6H and Figure 7C) ……………………………………*L*. (s. str.) *zayuana* sp. n.

### 4.2. Phenotypic Relationships Based on Aedeagi Shapes

In the present study, we analyzed the phallus shapes of all species of *Lyponia* s. str. (a total of 11 species) using GM analyses and further constructed their interspecific relationships using different methods (Figure 8). Given that any method has its own defects and limitations, we tried different methods to obtain more objective and dependable results. As expected, there is some discrepancy in the produced topologies, which likely resulted from different analysis data and optimal criteria of the methods [11]. However, most produced topologies (Figure 8A–C) demonstrate that closer relationships existed among the species of clade I (including *L. hainanensis*, *L. tamdaoensis*, *L. shaanxiensis*, *L. ruficeps* sp. n., *L. kuatunensis,* and *L. zayuana* sp. n.) and those of clade II (including *L. nepalensis*, *L. cangshanica*, *L. oswai*, *L. debilis,* and *L. delicatula*), which are supported by synapomorphy (Figure 8a,b). This is consistent with the results of the comparative morphology of phallus shape, which is either stout (Figure 7B–G) or slender (Figure 7H–L). Within clade I, a sister relationship of *L. hainanensis* + *L. tamdaoensis* is frequently recovered (Figure 8A–D) and supported by synapomorphy (Figure 8c). This finding is congruent with the GM analysis, and the two species are in closer positions in the first two relative warp axes (Figure 7). However, within clade II, *L. nepalensis* + *L. cangshanica* + *L. oswai* (Figure 8A–D) and *L. debilis* + *L. delicatula* (Figure 8A,C,E) are recovered as sister groups, separately, but they do not seem congruent with their relative shape variations of the phallus (Figure 7). Procrustes and Mahalanobis distances are used to capture shape variation, which is considered the best method for measuring shape differences among taxa [41,46,47,48,49,50]. So, they can effectively indicate phonetic relationships, summarizing overall patterns of similarity [41,51].

In constructing morphological phylogeny, male genitalia have been widely used in many arthropod groups by systematists [52], owing to its effective phylogenetic value in higher-level classifications [53,54,55,56,57,58,59]. Despite the opinion that genital evolution occurs too fast at the specific level to observe its phylogenetic inertia in the structures [60,61], this notion is contrary to the findings of Song and Bucheli [52] and most recently those of Liu et al. [11].

Unexpectedly, a gap still exists between our obtained results and the previous molecular phylogenic analysis [13]. In the study by Li et al. [13], five species of *Lyponia* s. str. were included in the analysis, and the phylogenetic relationship was recovered as (*L. delicatula* + *L. kuatunensis*) + (*L. oswai* + *L. shaanxiensis* + *L. debilis*). However, none of our produced topologies based on male genitalia shape are congruent with this result. Losos [62] argued that if rates of character evolution to speciation are relatively high, there may not be a relationship between the degree of the phylogenetic relationship and phenotypic similarity. If so, it is presumed that some non-genital characters of male [52] or female genitalia [63] are also involved in the speciation of *Lyponia*. Despite this, we should note that although molecular phylogenetic analysis has become the standard for inferring evolutionary relationships at present [64], the lack of samples of the representative species [e.g., 18] inevitably influenced the ability to elucidate the relationships among the species [65]. Nevertheless, more samples or loci are required in the future to reconstruct the phylogeny of *Lyponia* s. str.

## 5. Conclusions

In the present study, we review the lycid subgenus *Lyponia* s. str. and describe two new species from China, including *L. ruficeps* sp. n. and *L. zayuana* sp. n. Then, we investigate the phenotypic relationships among the species of *Lyponia* s. str. based on the phallus shapes with GM and PM analyses (using UPGMA, NJ and MP analyses). The produced topologies demonstrate that the species are usually divided into two clades (clade I and clade II) using UPGMA and NJ analyses based on the Procrustes and or Mahalanobis distances. Clade I is composed of six species (*L. hainanensis*, *L. tamdaoensis*, *L. ruficeps* sp. n., *L. shaanxiensis*, *L. kuatunensis,* and *L. zayuana* sp. n.) and is supported by a stout phallus (less than 3.6 times longer than wide). Clade II includes the remaining species (*L. nepalensis*, *L. cangshanica*, *L. oswai*, *L. debilis,* and *L. delicatula*) and is supported by a slender phallus (at least 4.1 times longer than wide). The sister groups, *L. hainanensis* + *L. tamdaoensis* (of clade I), *L. nepalensis* + *L. cangshanica* + *L. oswai*, and *L. debilis* + *L. delicatula* (of clade II) are frequently recovered. These results provide better understanding of the species diversity and evolution of *Lyponia* s. str. Nonetheless, more samples or other types of data are required in the future to verify the present results.

## Figures and Tables

**Figure 1 insects-15-00011-f001:**
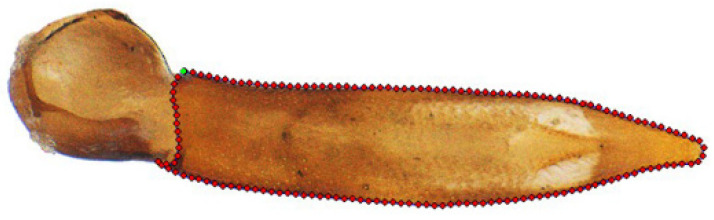
The curve used in the GM analysis as shown for *Lyponia* (s. str.) *ruficeps* sp. n.

**Figure 2 insects-15-00011-f002:**
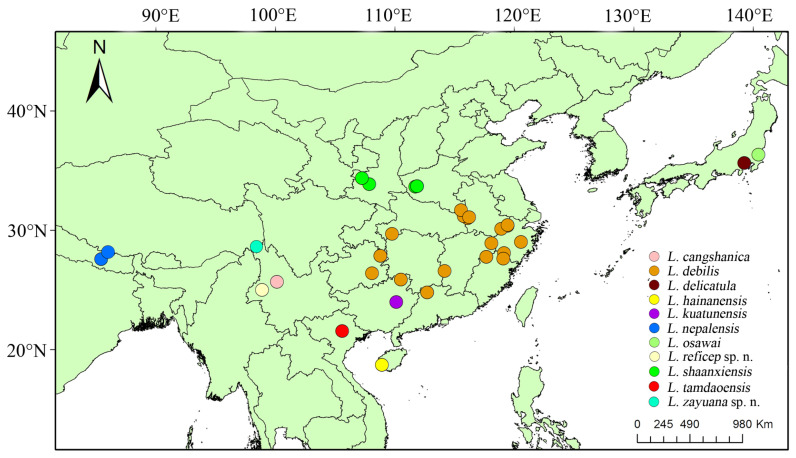
Distribution map of *Lyponia* s. str. in the world.

**Figure 3 insects-15-00011-f003:**
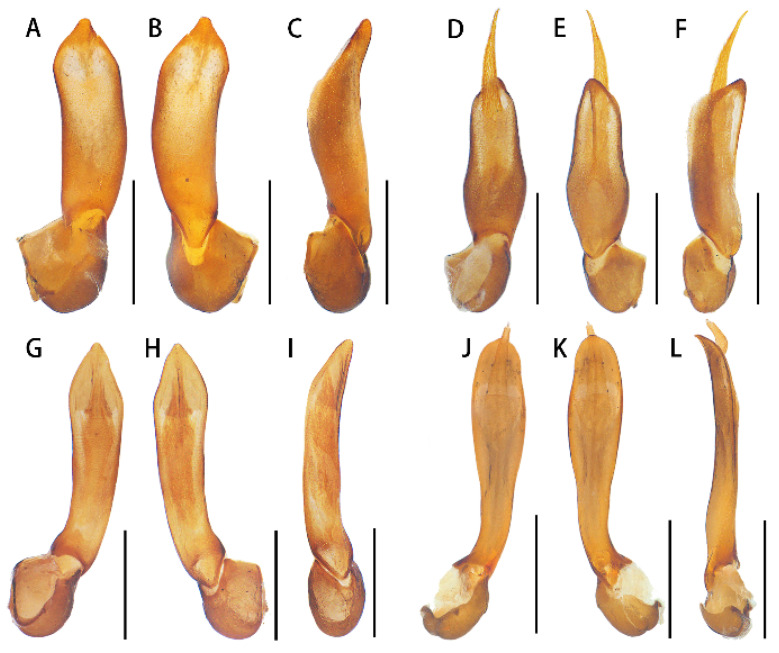
Aedeagi: (**A**–**C**). *Lyponia* (s. str.) *cangshanica* Li, Pang & Bocak, 2015; (**D**–**F**). *L.* (s. str.) *debilis* Waterhouse, 1878; (**G**–**I**). *L.* (s. str.) *kuatunensis* Bocak, 1999; (**J**–**L**). *L.* (s. str.) *hainanensis* Li, Bocak & Pang. (**A**,**D**,**G**,**J**)—ventrally views; (**B**,**E**,**H**,**K**)—dorsal views; (**C**,**F**,**I**,**L**)—lateral views. Scale bars: 0.5 mm.

**Figure 4 insects-15-00011-f004:**
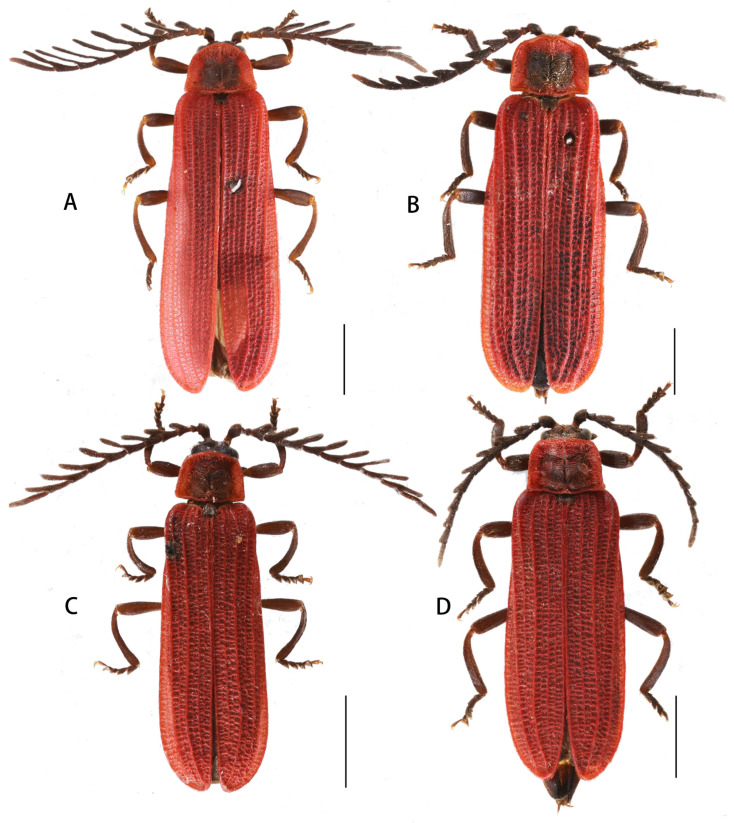
Habitus, dorsal view: (**A**,**B**). *Lyponia* (s. str.) *debilis* Waterhouse, 1878; (**C**,**D**). *L.* (s. str.) *kuatunensis* Bocak, 1999. (**A**,**C**). male; (**B**,**D**). female. Scale bars: 2.0 mm.

**Figure 5 insects-15-00011-f005:**
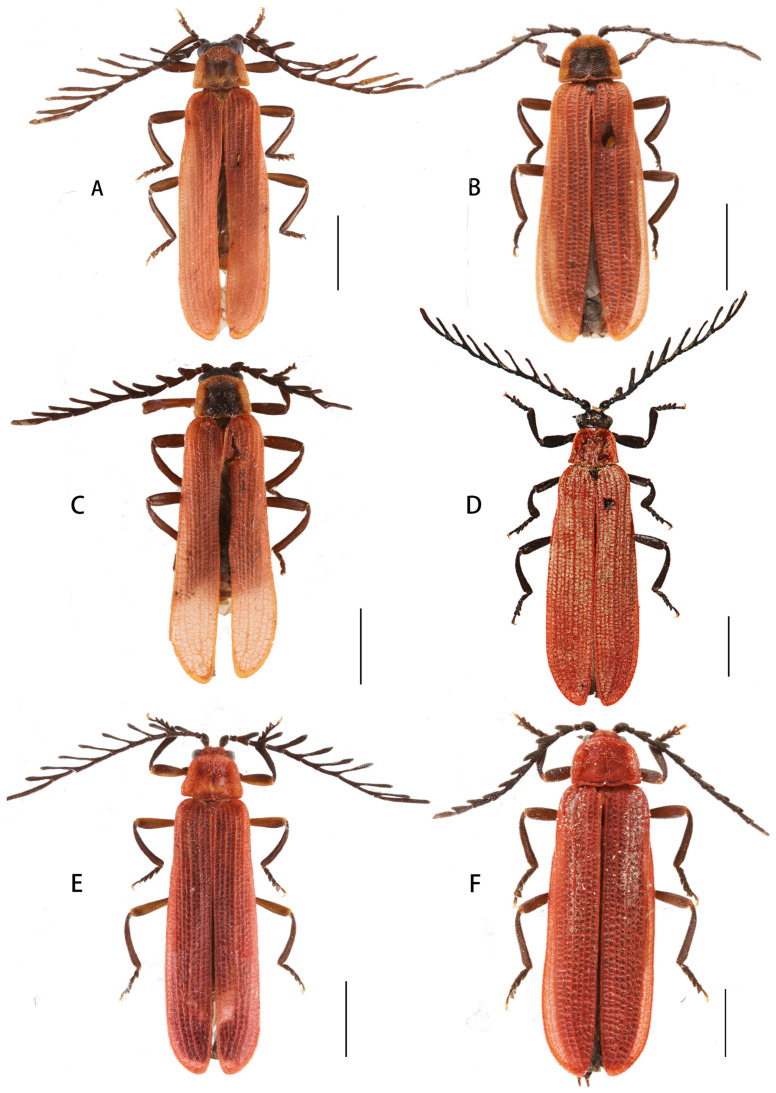
Habitus, dorsal view: (**A**,**B**). *Lyponia* (s. str.) *shaanxiensis* Kazantsev, 2002; (**C**). *L*. (s. str.) *nepalensis* Nakane, 1983; (**D**). *L.* (s. str.) *zayuana* sp. n.; (**E**,**F**). *L.* (s. str.) *ruficeps* sp. n. (**A**,**C**,**D**,**E**)**.** male; (**B**,**F**). female. Scale bars: 2.0 mm.

**Figure 6 insects-15-00011-f006:**
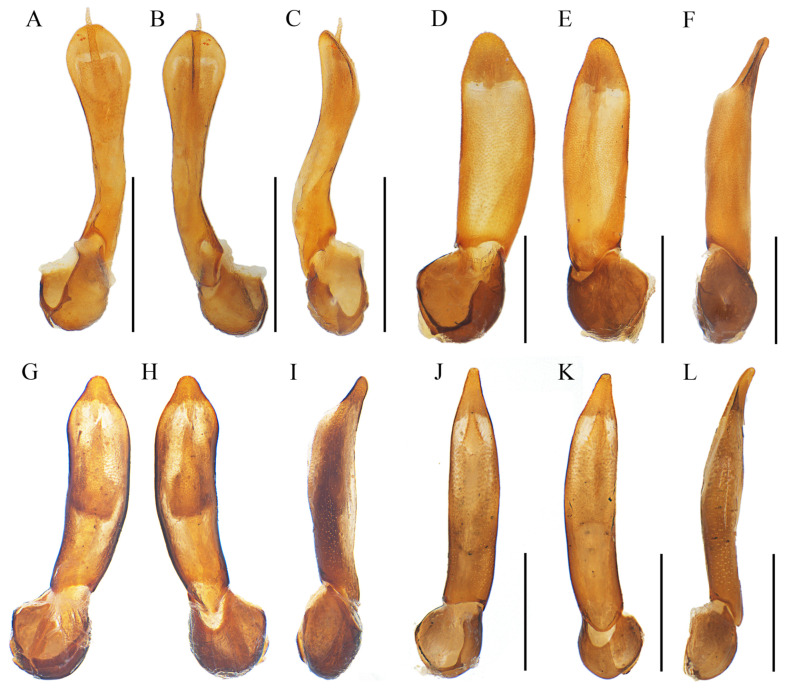
Aedeagi: (**A**–**C**). *Lyponia* (s. str.) *shaanxiensis* Kazantsev, 2002; (**D**–**F**). *L.* (s. str.) *nepalensis* Nakane, 1983; (**G**–**I**). *L.* (s. str.) *zayuana* sp. n; (**J**–**L**). *L.* (s. str.) *ruficeps* sp. n. (**A**,**D**,**G**,**J**)—ventrally; (**B**,**E**,**H**,**K**)—dorsally; (**C**,**F**,**I**,**L**)—laterally. Scale bars: 0.5 mm.

**Figure 7 insects-15-00011-f007:**
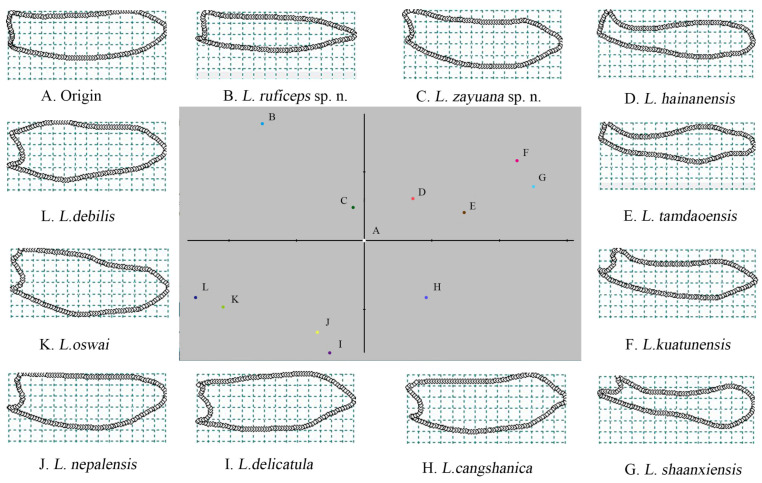
Shape differences in the phallus among the species of *Lyponia* s. str.: shape changes among species are implied by variations along the first two relative warp axes; (**A**–**L**) splines indicate deformation of the landmarks in comparison with the reference configuration.

**Figure 8 insects-15-00011-f008:**
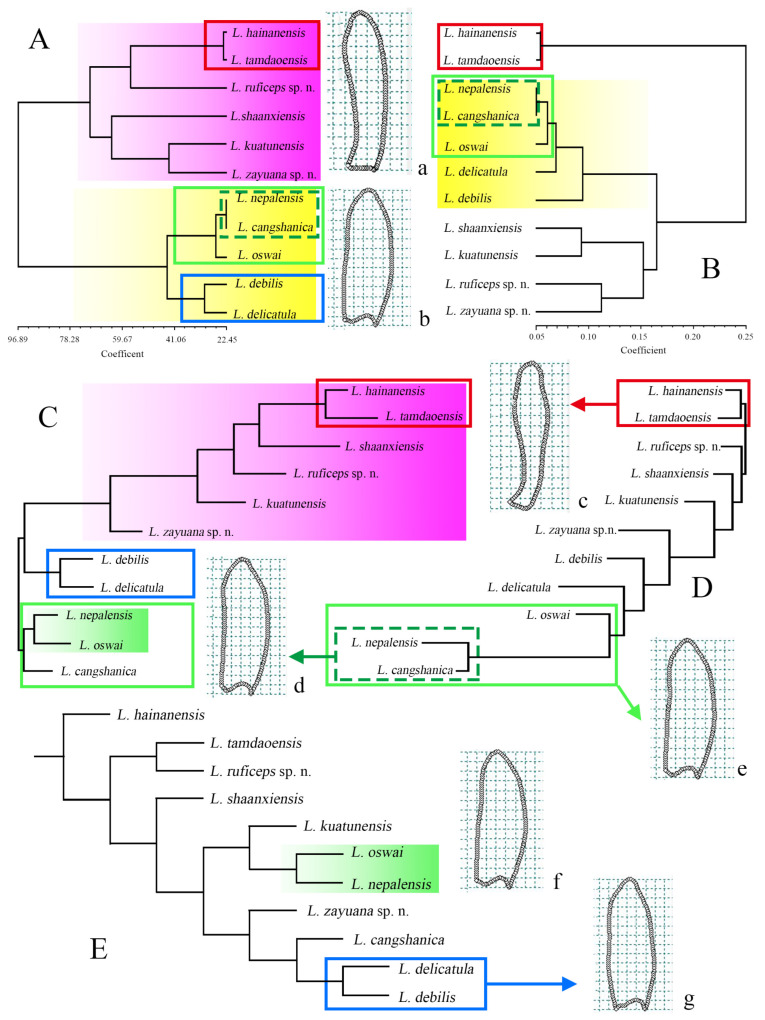
Comparisons of phenotypic relationships of *Lyponia* s. str.: (**A**,**B**) phylogenetic hypothesis based on Mahalanobis (**A**) and Procrustes (**B**) distances using UPGMA; (**C**,**D**) neighbor-joining tree for the species of *Lyponia* s. str. based on Mahalanobis (**C**) and Procrustes (**D**) distances with 1000 bootstrap replicates; (**E**) phylogenetic hypothesis based on two landmark configurations using MP analysis. The branches in different colors or dashed boxes represent the same clade recovered in different phenograms. The average shape of each branch is displayed in (**a**–**g**), respectively.

**Table 1 insects-15-00011-t001:** Information on the *Lyponia* s. str. species used in the GM analysis.

Species	Data Collection
*L. cangshanica* Li, Pang & Bocak, 2015	This study (China, Yunnan, Deqin)
*L. debilis* Waterhouse, 1878	This study (China, Zhejiang, Anji)
*L. delicatula* (Kiesenwetter, 1874)	Bocak [12]
*L. hainanensis* Li, Pang & Bocak, 2015	This study (China, Guangxi, Maoershan)
*L. kuatunensis* Bocak, 1999	This study (China, Guangxi, Damingshan)
*L. nepalensis* Nakane, 1983	This study (China, Xizang, Nyalam)
*L. osawai* Nakane, 1969	Bocak [12]
*L. shaanxiensis* Kazantsev, 2002	This study (China, Shaanxi, Ningshan)
*L. tamdaoensis* Kazantsev, 2002	Kazantsev [21]
*L. ruficeps* sp. n.	This study (China, Yunnan, Baoshan)
*L. zayuana* sp. n.	This study (China, Xizang, Zayu)

## Data Availability

All studied material is stored at the Institute of Zoology, Chinese Academy of Sciences, Beijing, China (IZAS) and Museum of Hebei University, Baoding, China (MHBU).

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
