# Peer review of "A Phylogenetic Morphometric Investigation of Interspecific Relationships of Lyponia s. str. (Coleoptera, Lycidae) Based on Male Genitalia Shapes"

_insects, 2023, doi:10.3390/insects15010011_

Round 1

Reviewer 1 Report

Comments and Suggestions for Authors

Abstract.

“Nominate”, but where? In the abstract they are not yet quoted

Usually it is better use the term aedeagus instead of phallus to avoid confusion to endophallus, the membranaceous internal sac of male genitalia. You quote the anatomical trait as aedeagus in row 556

Introduction.

The focus should be on the study taxon, and perhaps the more closely related groups. Please remove the out-of-topic parts.

The discussion about molecular and morphological data applied to phylogenetic analysys is redundant, since authors did not test here the differences, flaws and merits about the two types of data. A short phrase quoting both method will be more than enough.

The introduction could be safely reduced to the part from row 68.

The aim of the paper must be explicitly stated in the Introduction.

M&M

Row 109 - Some method was applied to minimize the risk to body parts malpositioning when measuring the total length?

Row 141 – which dataset was analized with Morphoj? The original coordinates dataset or the aligned one? Since authors stated that the points are semilandmarks, please conside that in Morphoj you cannot define points as semilandmarks. Thence, all the points are treated as true landmarks. This is an issue, and the method must be explained in details. If the wronfg dataset was used, the analyses must be done with the correct dataset (aligned specimens)

Figure 1 . The point configuration is perhaps too tick. Authors could do an Elliptic Fourier Analysis with a configuration such as the one shown in the figure, but the landmark approach is a bit different and doesn’t need a so detailed definition of the outline. Did the authors also tested configurations with less points?  How many true landmarks were included in the configuration? Usually they must be three al least. The other can be defined as semilandmarks in the sliders file.

Authors must define whether the coordinates dataset or the aligned one were analyzed in Past. Why the authors did not use the phylogenetic analysis within MorphoJ?

Results

The taxonomic part is allright, the identification keys are clear, with the alternative choices are sound.

The description of the species is clear and detailed. No changes are required.

All the figures are useful and of good quality

The GM analysis seems interesting, Once the critical points listed above will be properly addressed, any required change should be done.

Row 475 – configureurations (sic) – please check the manuscript for misspellings.

Discussion

The reason to include a second identification keys in this section is unclear. Please merge the two keys.

The Discussion must be checked, once the method issues will be examined.

Comments on the Quality of English Language

Please check the English text for misspellings, and some phrases in the Introduction could be modified.

Author Response

please see the response in the attachment, thanks.

Reviewer 2 Report

Comments and Suggestions for Authors

This is an interesting paper. The analysis of aedeagus shape breaks new ground. I have marked two places where wording should be altered, on the attached manuscript

Author Response

Please see the response in the attachment, thanks.

Reviewer 3 Report

Comments and Suggestions for Authors

Authors present an interesting usage of geometric morphometrics based on the shape of male genitalia, however, supplementary files were not available for reviewers. Which means I couldn't verify resultis of this manuscript. It is strange to me that editors allow to send this MS to reviewers without complete data. Nonetheless, my greates concern is about how authors shoose "clades". It is unclear which topology (method of phylogenetic reconstruction) is preferred by the authors. why "clades" are defined based on UPGMA method rather than MP analysis? Also it is not explained what color frames on the figure 8 mean. that should be clearly written in figure caption or somewhere within the text. Also I would appreciate more in depth discussion of these morpho results with published molecular hypotheses.

Author Response

please see the details in the attachment, thanks.

Round 2

Reviewer 1 Report

Comments and Suggestions for Authors

Authors made some changes and now the manuscript is surely improved. Hovewer, to maintain the first part of Introduction it is not a good choice. The readers are well-aware of the changes in the phylogenetic approach along years. Some indefinite references to phylogenetic methods is not really useful. If phylogenetic analysis's methods must be treated, they must be thorough evaluated and compared, examples must be given and merits and demerits of methods must be discussed. It should be better to give a detailed survey of the phylogenetic studies performend on these species. Furthermore, some comments were not supported by the correct references. e.g. rows 43-46 could benefit from some examples. Besides, there are many papers dealing with  morphological phylogenetic analyses based on large datasets of characters. Please check references.

When discussing the genitalia it should be better focus on the genitalia of the studied taxon, see rows 52-54. Insects is a quite large example.

Some misspelling are yet included in the manuscript e.g. row 171: phonetic instead of phenetic (although phylogenetic should even be better)

The Material & Methods section was improved, and now the used methods are more clearly described.

The position of the identification keys in the Discussion is not correct. If authors have generated these identification keys they must be placed in the Results, also in a section of its own could do. If the keys were estrapolated from another paper, and here modified: they could be at least included in the Supplementary Materials. In the Discussion section only the results gained by the present analyses must be discussed. Thence, the authors could briefly describe the structures of antennae and so on, and refer to the keys for any further information. The Discussion is not the ordinary position of the identification keys, please check the common format of taxonomic papers.

Some of these similarities found in the present manuscript are not a problem, as (e.g.) the legends of the figures: in both papers there are groups of images defined by the same series of letters. It is likely purely coincidental.

Different are the cases from Introduction, Material & Methods and so on. There the authors included many phrases from another paper of them without refer to it. This is really strange since the paper was quoted at one point as reference 16. Nevertheless, before this quotation there are many phrases almost identical in the two papers deserving a more correct attribution. Although this could not be strictly considered a plagiarism, authors wrote both papers, and some changes must be done. Surely authors should make an effort to paraphrase the sentences changing the text.

In the Discussion section authors could merely quote the former paper without borrowing from themselves. and discuss the results gained from the present research.

Although the overlaps seem more due to careless than other reasons, I could suggest to the authors to rewrite the marked parts. Some of  the overlapping phrases could be easily omitted without affecting the soundness of the manuscript. 

Comments on the Quality of English Language

In the text some errors must be mended

Author Response

Please see the details in the attachment, thanks.

Reviewer 3 Report

Comments and Suggestions for Authors

The manuscript has been imroved. In my opinion the word "phallus" should be replaced by for example "aedeagus".

Comments on the Quality of English Language

Although I am not a native speaker some parts of the text should be linguistically checked. For example "Figure 8. Comparing phenotypic relationships of Lyponia s. str" should be "Comparison" not "Comparing".

Author Response

(The authors gave the same response as above.)
